# Effects of Exogenous Ergothioneine on *Brassica rapa* Clubroot Development Revealed by Transcriptomic Analysis

**DOI:** 10.3390/ijms24076380

**Published:** 2023-03-28

**Authors:** Yuting Zhang, Guizhu Cao, Xiaonan Li, Zhongyun Piao

**Affiliations:** Molecular Biology of Vegetable Laboratory, College of Horticulture, Shenyang Agricultural University, Shenyang 110866, China

**Keywords:** Chinese cabbage, *Plasmodiophora brassicae*, ergothioneine, transcriptome, phenylpropanoid biosynthesis

## Abstract

Clubroot disease is a soil-borne disease caused by *Plasmodiophora brassicae* that leads to a serious yield reduction in cruciferous plants. In this study, ergothioneine (EGT) was used to culture *P. brassicae* resting spores, the germination of which was significantly inhibited. Further exogenous application of EGT and *P. brassicae* inoculation in Chinese cabbage showed that EGT promoted root growth and significantly reduced the incidence rate and disease index. To further explore the mechanism by which EGT improves the resistance of Chinese cabbage to clubroot, a Chinese cabbage inbred line BJN3-2 susceptible to clubroot treated with EGT was inoculated, and a transcriptome analysis was conducted. The transcriptome sequencing analysis showed that the differentially expressed genes induced by EGT were significantly enriched in the phenylpropanoid biosynthetic pathway, and the genes encoding related enzymes involved in lignin synthesis were upregulated. qRT-PCR, peroxidase activity, lignin and flavonoid content determination showed that EGT promoted the lignin and flavonoid synthesis of Chinese cabbage and improved its resistance to clubroot. This study provides a new insight for the comprehensive prevention and control of cruciferous clubroot and for further study of the effects of EGT on clubroot disease.

## 1. Introduction

*Plasmodiophora brassicae*, a soil-borne, obligate, biotrophic pathogen causing clubroot, causes serious damage to the yield and quality of cruciferous crops [1]. Infection by *P. brassicae* restricts the transport of nutrients and water, hindering plant development and resulting in plant withering and even death [2]. The ability of *P. brassicae* to survive in the soil for a long time and reproduce on a variety of host plants under a variety of ecological and climatic conditions, and its genetic diversity, poses great challenges for its control [3]. Currently, there are many genetic studies on clubroot resistance (CR) in the genus *Brassica*. Therefore, a better understanding of the mechanisms of clubroot disease development and host–*P. brassicae* interactions may provide strategies for improving plant tolerance to *P. brassicae* infection and establishing a theoretical basis for breeding resistant varieties.

Cell walls are dynamically developed in different parts of the plant and are regulated and remodeled to varying degrees in response to cell growth, elongation, and pathogen attack [4]. *Plasmodiophora brassicae* infects root hairs to form root nodules, resulting in the abnormal division and expansion of host root cells [5]. *P. brassicae* infection affects cell wall-related biological processes, in which the synthesis of cellulose, pectin, and lignin is downregulated in processes related to cell wall stability and firmness, while cell wall-degrading enzymes are upregulated [6]. *P. brassicae* reduces the structural stability of the host cell wall and increases its elasticity and extensibility, so that *P. brassicae* can grow and develop better and produce more dormant spores [6]. Lignin is a non-degradable mechanical barrier for most microorganisms, and host plants can increase their resistance by increasing lignification to resist pathogen attack [7,8]. Lignin deposition in infected cells prevents the spread of toxins and enzymes from the pathogen to the host, as well as prevents the transfer of water and nutrients from the host cells to the pathogen [9]. High levels of cell wall lignification provide an effective defense for plants against pathogens [10].

Currently, there are many studies focused on the pathogenic mechanism of *P. brassicae*, but the understanding of its pathogenic mechanism is still incomplete. With the in-depth study of host–*P. brassicae* interactions, the transcriptional changes after *P. brassicae* infects a host plant have been revealed. Flavonoids and lignin have many physiological functions and play multiple roles in plant disease resistance. Transcriptome analysis has shown that the expression of genes involved in lignin biosynthesis increases in both susceptible and resistant varieties after infection, and genes involved in the phenylpropanoid biosynthesis pathway are upregulated earlier and at a higher level in resistant plants than in susceptible plants [11]. In the early stage of *P. brassicae* infection in *Arabidopsis thaliana*, the biosynthesis pathways of flavonoids and lignin are significantly enhanced [12]. In the late stage of *P. brassicae* infection, *A. thaliana* produces important differential metabolites, such as flavonoids and lignin, which play an important role in clubroot resistance [13]. Naringenin, quercetin, and kaempferol are significantly accumulated in *A. thaliana* roots infected with *P. brassicae* [14].

Ergothioneine (EGT) is a rare natural chiral amino acid that widely exists in animals and plants [15]. EGT has attracted much attention due to its unique antioxidant and cytoprotective properties. EGT prevents the formation of free radicals, such as OH, and scavenges free radicals, such as hypochlorous acid (HClO), peroxynitrite, and reactive oxygen species (ROS) [16,17,18,19,20]. EGT interacts with other natural antioxidant defense systems in vivo to activate the intracellular antioxidant pathway of MAPKs and to regulate the levels of peroxidases and superoxide dismutase [21,22,23]. Due to EGT’s antioxidant properties, it plays a powerful cytoprotective role in some important cells and tissues [24]. EGT inhibits the peroxidation of a mixture of H_2_O_2_ and hemoglobin in arachidonic acid [16]. Moreover, EGT can inhibit nitrite-induced oxyhemoglobin oxidation by scavenging nitric oxide, which retards or reverses the formation of iron-bound hemoglobin and methemoglobin [25,26]. Medically, EGT deficiency has been shown to increase the risk of human disease. Lower blood EGT levels are associated with the morbidity of Parkinson’s disease (PD) [27], mild cognitive impairment (MCI) [28], Crohn’s disease (CD) [29], and frailty [30], while correspondingly higher blood EGT levels are associated with a lower risk of cardiometabolic disorders and associated mortality [31] and a lower prevalence of peripheral neuropathy [32]. However, few functional studies of EGT in response to biotic stress have been reported.

In this study, Chinese cabbage was inoculated with *P. brassicae* after the exogenous application of EGT. To explore the role of exogenous EGT in the interaction between Chinese cabbage and clubroot, root morphological observation, physiological index determination, clubroot incidence investigation, transcriptome analysis, lignin and flavonoid content detection, key enzyme activity detection, and phenylpropanoid pathway key gene verification were performed. The results provide an important theoretical basis for the prevention and control of clubroot.

## 2. Results

### 2.1. Effects of Different EGT Concentrations on the Activity of the Resting Spores of P. brassicae

The root exudates of cruciferous plants promote the germination of the resting spores of *P. brassicae*. After 1% orcein staining for 15 s, the ungerminated resting spores were stained red, while the germinated resting spores were not stained (Appendix A). At 48, 72, and 96 h, the germination rate of resting spores in the treatment group with root exudates was significantly higher than that of resting spores in the control group (treated with water). The germination rate of resting spores was significantly decreased when they were cultured with 0.1 mM of EGT and root exudates for 72 and 96 h, compared with the control group treated only with root exudates (Appendix A). The germination rate of resting spores increased with time, while the EGT treatment significantly inhibited the germination rate of resting spores in root exudates.

### 2.2. Effect of EGT on Chinese Cabbage Infected with P. brassicae

‘BJN3-2′ inoculated with only Pb4 (Pb4 group) did not form clumps at 10 d but did form small tumors at 14 d and large tumors at 19 and 27 d (Figure 1A). Exogenous application of 0.05 mM of EGT for 24 h, followed by inoculation with Pb4 (EPb group), increased the resistance of Chinese cabbage to clubroot (Figure 1A). No nodule formation was observed in the EPb group at 10 days post-inoculation (dpi). The disease rate (DR) was 83% in the Pb4 group at 14 dpi, while the DR in the EPb group was significantly reduced to 33% (Figure 1B). The disease index (DI) of the EPb group was significantly reduced from 14 to 27 dpi (Figure 1C). The relative content of *P. brassicae* was higher in the Pb4 group than in the control group at 10 dpi, significantly higher in the Pb4 group than in the control group and EPb group from 14 to 27 dpi, and higher in the EPb group than in the control group (Figure 1D). Exogenous EGT significantly decreased the DR and DI of Chinese cabbage clubroot.

To study the effect of exogenous EGT on the root growth of Chinese cabbage, the root length, root volume, and root activity of Chinese cabbage were measured. Compared with the control group, exogenous EGT treatment increased the root length, root volume, and root activity of Chinese cabbage (Figure 2a–c). The root length of the treatment group was significantly higher than that of the control group at 10 and 14 d, and the root length of the EPb group was higher than that of the other groups at 10 d (Figure 2a). Without the stress of *P. brassicae*, exogenous EGT promoted an increase in root volume. The root volume increased significantly after inoculation, while the root volume of the inoculated roots was inhibited after exogenous EGT treatment (Figure 2b). The root activity of the EGT group was significantly higher than that of the control group. Furthermore, the root activity was lower in the EPb group than in the Pb4 group. The root activity of the Pb4 and EPb groups was significantly lower than that of the control and EGT group at 27 d (Figure 2c). Therefore, EGT promoted the root growth of Chinese cabbage and improved its resistance to clubroot.

### 2.3. DEG Analysis of Exogenous EGT Treatment on Chinese Cabbage Root Development

To determine the effect of exogenous EGT, root samples were collected from the control, EGT, Pb4, and EPb groups at 0, 10, 14, 19, and 27 dpi for transcriptome sequencing. RNA-seq data were paired-ends captured. The sequencing read length was 150 bp. The average raw read number was 46,270,556 for each sample (Appendix A). On average, 36,243,221 reads per sample were mapped to the *B. rapa* genome (Appendix A). Three biological replicates were used to detect minor differential gene expression among treatments, and the time points were used to assess our data quality by carrying out PCA. The results of the principal component analysis (PCA) showed that the variance among the three replicates in each group was small, and the sample repeatability was high and reliable. Thus, the data could be used for further analysis (Figure 3a,b). To study the changes in the transcription level in Chinese cabbage under EGT treatment, the differentially expressed genes (DEGs) between EGT and CK were analyzed 10 days after inoculation. Of 845 DEGs, 594 were upregulated and 251 were downregulated (Figure 3c). The Kyoto Encyclopedia of Genes and Genomes (KEGG) enrichment analysis indicated that phenylpropanoid biosynthesis, nitrogen metabolism, and flavonoid biosynthesis were significantly enriched (Figure 3d). The phenylpropanoid metabolic pathway of Chinese cabbage was significantly activated by exogenous EGT.

### 2.4. DEG Analysis of Chinese Cabbage in Response to P. brassicae Infection

To study the effect of EGT on the interaction between Chinese cabbage and P. brassicae, the DEGs between EPb and Pb4 at 10, 14, 19, and 27 days were analyzed. Most DEGs were observed at 10 dpi, with a total of 2186 DEGs, of which 1730 were upregulated and 456 were downregulated. At 14, 19, and 27 d, there were 2385, 245, and 1829 DEGs, respectively (Figure 4a). The EGT treatment resulted in dramatic changes in gene expression. To analyze the effect of EGT at the early stage of clubroot infection, the transcription levels of differential genes in the control, EGT, Pb4, and EPb groups at 10 and 14 dpi were analyzed and presented in a Venn diagram. As shown in Figure 5b, the gene set composed of 120 common differential genes (circled in red box) was selected for further analysis (Figure 4b, Appendix A). The KEGG enrichment analysis of 120 common differential genes showed that phenylpropanoid biosynthesis was significantly enriched, followed by fatty acid degradation and nitrogen metabolism. Therefore, we further analyzed the phenylpropanoid pathway (Figure 4c).

To further explore the core disease resistance genes, we performed a weighted gene co-expression network analysis (WGCNA) of all genes from the transcriptome. Each row and column of the heat map represents a gene that describes the connection relationship and the coverage relationship of the topological network. The darker the color, the higher the degree of connection. The different genes were assigned to 12 different modules and marked with different colors (Appendix A). A core network was identified as a critical module for disease resistance (brown module), and genes with known functions were selected (Figure 5a,b). Each node represents a gene, and the connection lines (edges) between genes represent the correlation of gene co-expression, revealing the key modules. The core gene of the brown module was the lignin synthase gene CCR (Figure 5c). CCR, the first key enzyme in the shift of lignin from the common phenylpropanoid pathway to a specific synthetic pathway, catalyzes the reduction in the CoA esters of three hydroxycinnamic acids to the corresponding cinnamaldehydes.

### 2.5. Effect of EGT on Lignin and Flavonoid Content

The transcriptome gene enrichment analysis and WGCNA showed that genes were mainly enriched in the phenylpropanoid pathway. Thus, we detected the lignin and total flavonoid content. The lignin content of the EPb group was lower than that of the other groups at 10 d. However, at 14 d, the lignin content of the EPb group was higher than that of the other groups. The EGT and Pb4 groups had a higher lignin content than the control group. The lignin content of the inoculated group was higher at 19 d. At 27 d, the lignin content of the EGT group was higher than that of the control group, and it was higher in the EPb group than in the Pb4 group (Figure 6a). The flavonoid content differed in the early stage of treatment. The flavonoid content increased significantly at 10 and 14 dpi, and in the Pb4 group compared to the control and EGT groups (Figure 6b).

### 2.6. Effect of EGT on the H_2_O_2_, MDA, and POD Content in the Roots

The hydrogen peroxide (H_2_O_2_) content in the roots increased and then decreased. The H_2_O_2_ content of the EPb group was 1.27 and 1.49 µmol/g at 10 and 14 dpi, respectively, which was significantly higher than that of the Pb4 group. The inoculated group had a significantly higher H_2_O_2_ content than the non-inoculated group. The H_2_O_2_ content in the EPb group decreased at 19 dpi (Figure 6c). Exogenous EGT decreased the malondialdehyde (MDA) content. The MDA content in the EGT group was lower than that in the control group at 10 and 14 d, decreasing by 18.46 and 25.57%, respectively. The MDA content was highest in the Pb4 group and increased continuously over time. The content of MDA in the EPb group was lower than that in the Pb4 group, which decreased by 45.08%, 48.74%, 19.69%, and 30.34% from 10 d to 27 d, respectively (Figure 6d). The POD content in the EPb group was higher than that in the Pb4 group at 10 and 14 d. There was no significant difference in POD activity between the treatment groups and the control group at 10, 14, and 19 d. The POD activity in the treatment group was higher than that in the control group at 27 d, and there was no significant difference between the treatment groups (Figure 6e).

### 2.7. Quantitative RT-PCR Verification of the Phenylpropanoid Pathway Genes

EGT treatment activates the phenylpropanoid biosynthetic pathway. To verify the accuracy of the transcriptome, six genes were randomly selected in the phenylpropanoid biosynthetic pathway for qRT-PCR validation, and the expression levels of the genes were consistent with the transcriptome data. The core genes for phenylalanine metabolism, namely phenylalanine ammonia-lyase (*PAL*), trans-cinnamate 4-monooxygenase (*C4H*), 4-coumarate-CoA ligase (*4CL*), caffeoyl-CoA O-methyltransferase (*CCOAOMT1*), flavonol synthase (*FLS*), and anthocyanidin synthase (*ANS*), were more highly expressed after the application of exogenous EGT, and the difference in expression was the largest at 14 dpi (Figure 7). The genes exhibited a similar expression trend, indicating that the expression data obtained by RNA sequencing in the present study were reliable (Appendix A). Exogenous EGT enhanced the resistance of Chinese cabbage to clubroot by promoting the expression of phenylpropanoid pathway genes.

## 3. Discussion

Root exudates can stimulate the germination of resting spores, and this stimulation is closely related to the source of root exudates. Further, the root exudates of cruciferous host crops are more effective than those of non-host plants [33].Through a comparative analysis of the germination rates of different treatment groups, we found that the resting spores of *P. brassicae* germinated in water, but the germination rate was low. Compared with the water control, the root exudates of Chinese cabbage significantly increased the germination rate of resting spores. The germination rate of *P. brassicae* resting spores was consistent with the results of previous studies, which confirmed the conclusion that the root exudates of Chinese cabbage promoted the germination of *P. brassicae* dormant spores [34]. Compared with the control, root exudates plus EGT significantly inhibited the germination of *P. brassicae* dormant spores, but there was no concentration-dependent effect of EGT treatment (Appendix A). Due to the limited research on ergothioneine, the reason for the effect of ergothioneine on the germination of *P. brassicae* could not yet be inferred. The results showed that exogenous EGT significantly reduced the disease rate and index of clubroot and improved the resistance of Chinese cabbage to clubroot (Figure 1B,C). However, it is not clear which genes are responsible for EGT synthesis in *B. rapa*. In the future, it is necessary to explore the synthetic genes of EGT and try to improve the resistance of Brassica plants to clubroot by increasing the synthesis of endogenous EGT.

The results of phenotypic observation, statistics of DR and DI, and analysis of *P. brassicae* content in Chinese cabbage showed that exogenous EGT treatment promoted root length, root volume, and root activity and reduced the DR and DI of Chinese cabbage clubroot. To the best of our knowledge, this is the first time that the effects of EGT on promoting root growth and responding to *P. brassicae* stress has been revealed. As shown in Figure 1, compared with the Pb4 group, DI and *P. brassicae* DNA quantity of plants treated with EGT (Pb4 + EGT group) were significantly decreased at 14 d, 19 d, and 27 d, although the disease rate was 100% until 27 d. There was no significant difference in biomass from 14 d to 27 d, only a slight yellowing phenomenon at the edge of the infected Chinese cabbage leaves in Pb4 group. *P. brassicae* infection limits the transport of nutrients and water in Chinese cabbage and hinders its growth and development. When harvesting in the field, we can observe that the yield of Chinese cabbage infected by *P. brassicae* is significantly reduced. However, in this experiment, plants were grown in plug trays for each treatment and cultured in the greenhouse. Thus, further field trials are needed to measure the biomass during Chinese cabbage heading harvest time.

H_2_O_2_ is an important signal molecule in plants, and the rapid increase in endogenous H_2_O_2_ in the early stage of stress leads to the expression of stress-related genes [35,36,37]. The same conclusion was obtained in this experiment; the content of H_2_O_2_ increased significantly at 10 and 14 dpi. As the end product of membrane lipid peroxidation, MDA can represent the degree of cell membrane lipid peroxidation and plant stress injury, and the MDA content increases when plants are stressed [38]. In this study, the MDA content was significantly increased after *P. brassicae* inoculation, while it was significantly decreased after exogenous EGT application, indicating that EGT may inhibit MDA accumulation and improve the antioxidant capacity to alleviate the damage caused by *P. brassicae* to Chinese cabbage and improve disease resistance. POD is a key enzyme in the enzymatic defense system of plants under adverse conditions [38]. In this experiment, the peroxidase content in the EPb group was higher than that in the Pb4 group at the early stage of inoculation, but it was not significant. Therefore, we speculated that EGT might not be involved in enhancing the resistance of Chinese cabbage to clubroot by increasing POD activity.

The KEGG enrichment analysis showed that the phenylpropanoid biological pathway was significantly enriched. The annotation analysis of the genes enriched in the pathway showed that exogenous EGT activated the phenylpropanoid biosynthesis pathway and that most genes were involved in the biosynthesis of lignin. Mutations in the key lignin biosynthesis gene, cinnamoyl-CoA reductase (*CCR*), affect lignin synthesis, which in turn alters physical properties, such as secondary cell wall strength and stiffness [39]. In tobacco, the reduction in *CCR* activity affects the amount of lignin synthesis and changes the proportion of lignin monomers, and the line with the most reduced *CCR* activity also has the most reduced lignin synthesis content [40]. Caffeoyl coenzyme methyltransferase (*CCoAOMT*), which plays an important role in lignin synthesis, is downregulated in susceptible oilseed rape lines [41]. 4-Coumarate-CoA ligase (*4CL1*), cinnamoyl-CoA reductase (*CCR1*), and cinnamyl alcohol dehydrogenase (*CAD5*) genes involved in lignin biosynthesis in *Arabidopsis* were downregulated at 4 dpi [42]. The infection of *Brassica oleracea* with clubroot affects processes associated with the cell wall, with substances associated with cell wall stability and rigidity (cellulose, pectin, or lignin synthesis), which are downregulated, and cell wall-degrading enzymes (expansins) of host cell wall flexibility, which are upregulated [6]. Six genes involved in lignin biosynthesis in the biosynthetic pathway are expressed at higher levels in resistant plants than in susceptible plants [43]. Previous studies have shown that cell wall lignification provides the host with a robust defense against fungal infection [10]. In other words, the reduced expression of genes involved in lignin biosynthesis leads to a decrease in lignin content in the cell wall, which fails to play a better defensive role in the process of *P. brassicae* infection, making plants more susceptible to infection and leading to larger nodule formation and severe disease. The determination of lignin and flavonoid content confirmed that exogenous EGT promoted lignin and flavonoid biosynthesis during the formation of root nodules in the early stage of infection. The induction of EGT activated the phenylpropanoid biosynthesis pathway in host plants, led to the upregulated expression of related genes, promoted lignin and flavonoid synthesis, and enhanced the disease resistance of host plants.

In conclusion, exogenous EGT inhibited the germination of the resting spores of *P. brassicae*, promoted root growth, increased the H_2_O_2_ content, decreased the MDA content, and reduced the degree of membrane lipid peroxidation. By activating the phenylpropanoid biosynthesis pathway, exogenous EGT increased the expression of related genes, promoted the synthesis of lignin and flavonoids, and improved the resistance of Chinese cabbage to clubroot. This study revealed a new function of EGT in promoting root growth and resistance to clubroot. It provides an important theoretical basis for improving clubroot resistance. In the future, we will try to apply EGT treatment in the field to control clubroot.

## 4. Materials and Methods

### 4.1. Plant Materials and Pathogen Inoculation

The Chinese cabbage inbred line BJN3-2 was used as the experimental material. The JSLYG-139 strain was used in this experiment and was identified to be *P. brassicae* pathotype Pb4 based on the SCD classification system [44].

### 4.2. Effects of Different EGT Concentrations on the Resting Spores of P. brassicae

To study the effect of EGT (Haoyuan Chemexpress Co., Ltd., Shanghai, China) on the germination of *P. brassicae* resting spores, the seeds of BJN3-2 were washed with 75% alcohol for 30 s, 10% NaClO for 1 min, 2% NaClO for 10 min, and sterile distilled water, and then sown in MS medium [45]. After 7 days of culture, Chinese cabbage seedlings were transferred to 1/2 MS liquid medium, and root exudates were collected and stored at −80 °C. Sterile water and sterile water plus 0.05 mM of EGT were set as blank and negative controls, respectively. Root exudates were supplemented with 0.05 and 0.1 mM of EGT, respectively, as experimental groups. The mixtures with different treatments were put into conical flasks and placed on a shaker under dark conditions for shaking culture (130 rpm, 24 °C), then observed at 0 h, 24 h, 48 h, and 72 h. The preparation of *P. brassicae* resting spores was performed following the method of Ma et al. [46]. *P. brassicae* resting spores (1 × 10^8^ spores·mL^−1^) were mixed with root exudates for culture. The resting spores of *P. brassicae* were stained with orcein, while the germinated spores were not stained [47].

### 4.3. Exogenous EGT Treatment

The 29-day-old BJN3-2 was injected with 1 mL of 0.05 mML^−1^ EGT or distilled water (negative control) into the soil near the root system with a medical syringe, then inoculated with 1 mL of *P. brassicae* (1 × 10^8^ spores·mL^−1^) 24 h later. Chinese cabbage seedlings were grown under LD conditions (16 h light/8 h dark photoperiod) in the presence of cool-white fluorescence light at 22 ± 1 °C at a relative humidity of 65–70%. Three biological replicates were conducted for each treatment, and each replication contained 30 plants. Roots were collected at five time points (0, 10, 14, 19, and 27 d) for transcriptome sequencing.

### 4.4. Disease Assessment

BJN3-2 root infection was assessed on a 0–4 scale: 0, no symptoms; 1, small galls on the lateral roots; 2, larger galls on the lateral roots; 3, small galls on the main roots; and 4, severe galling of tissues of both lateral and main roots. The DI was calculated according to the following formula: DI = [nw] × 100/4T, where n is the number of plants in each class, w is disease symptoms (0–4), and T is the total number of plants tested [48,49].

### 4.5. Detection of Root Physiological Indices and Activity

The root length and volume of Chinese cabbage were determined using a Microtek ScanWizard EZ scanner and SC-E image analysis software (Hangzhou Wanshen Detection Technology Co., Ltd., Hangzhou, China). Root activity was determined using the triphenyl tetrazolium chloride (TTC) reduction method [50,51].

### 4.6. RNA Extraction and Transcriptome Sequencing

Total RNA was extracted from the CK (0, 10 d) and EGT, Pb4, and EPb samples at four time points (10, 14, 19, and 27 d) using Trizol reagent (Invitrogen Life Technologies, Carlsbad, CA, USA), according to the manufacturer’s instructions. The concentration, quality, and integrity were determined using a NanoDrop spectrophotometer (Thermo Scientific, Waltham, MA, USA). mRNA was purified from total RNA using poly-T oligo-attached magnetic beads. Fragmentation was performed using divalent cations at an elevated temperature in an Illumina proprietary fragmentation buffer. First-strand cDNA was synthesized using random oligonucleotides and SuperScript II. Second-strand cDNA synthesis was subsequently performed using DNA Polymerase I and RNase H. The remaining overhangs were converted into blunt ends via exonuclease/polymerase activities, and the enzymes were removed. After adenylation of the 3′ ends of the DNA fragments, Illumina PE adapter oligonucleotides were ligated to prepare for hybridization. To select cDNA fragments of the preferred 200 bp length, the library fragments were purified using the AMPure XP system (Beckman Coulter, Beverly, CA, USA). The sequencing library was then sequenced on a Hiseq platform (Illumina, San Diego, CA, USA) by Shanghai Personal Biotechnology Cp. Ltd. (Shanghai, China).

### 4.7. Identification of Differentially Expressed Genes and Validation of RNA Sequencing by qRT-PCR

A DEG analysis was performed using DESEQ [52] to screen the DEGs with log2foldchange > 1 and a false discovery rate (FDR) < 0.05. Gene annotation was performed using the Kyoto Encyclopedia of Genes and Genomes (http://www.kegg.jp/ (accessed on 18 September 2022)).

To validate the RNA sequencing results, six genes were randomly selected from the phenylpropanoid metabolic pathway for qRT-PCR validation. cDNA was synthesized from total RNA using the PrimeScript™ RT Reagent Kit with gDNA Eraser (Perfect Real Time) (TaKaRa, Beijing, China), according to the manufacturer’s instructions. This was used as a template for PCRs, which were conducted on a Veriti Thermal Cycler (Thermo Fisher Scientific, Waltham, MA, USA). Relative expression was calculated according to the 2^–ΔΔCT^ method [53]. Each assay was repeated at least three times. The primers for qRT-PCR are listed in Appendix A.

### 4.8. Construction of Gene Co-Expression Networks and Prediction of Key Genes

Co-expression networks were constructed using the weighted correlation network analysis (WGCNA) package in R (v3.4.2). Based on the level of gene expression, the TOMSimilarity module was used to calculate the co-expression similarity coefficient between genes, and the pickSoftThreshold function of the software package was used to select parameters and perform a weighted calculation. Combined with the biological characteristics of the sample, the expression profile characteristics of module characteristic genes and the modules related to biological characteristics were screened. Cytoscape_v3.8.0 software was used to draw the gene network map for each module.

### 4.9. Determination of Lignin, Total Flavonoid, H_2_O_2_, MDA, and POD Content

H_2_O_2_, MDA, POD, and total flavonoid content were detected using a plant flavonoid kit, a peroxidase kit, an MDA kit, and an H_2_O_2_ kit (Michybio Company, Suzhou, China). The lignin content was determined following the method of Zhang et al. [54].

## Figures and Tables

**Figure 1 ijms-24-06380-f001:**
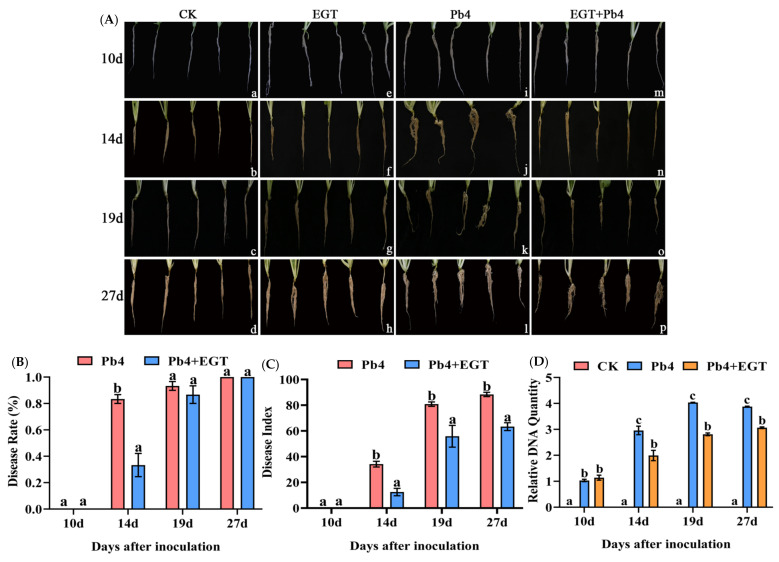
Effects of exogenous EGT treatment on root growth and disease development of Chinese cabbage. (**A**) Observation of root phenotype. (**a**–**d**): BJN3-2 at 10, 14, 19, and 27 d; (**e**–**h**): BJN3-2 with EGT at 10, 14, 19, and 27 d; (**i**–**l**): BJN3-2 with Pb4 at 10, 14, 19, and 27 d; (**m**–**p**): BJN3-2 inoculated with Pb4 after application with exogenous ergothioneine at 10, 14, 19, and 27 d. (**B**,**C**) Disease rate and disease index of clubroot. The red and blue bars represent the Pb4 group (inoculated with Pb4) and the EPb group (inoculated with Pb4 after exogenous applications of EGT), respectively. (**D**) Relative quantitative detection of *P. brassicae* content. Values were log10 transformed. Red, blue, and yellow bars represent the control, Pb4, and EPb groups, respectively. Black letters indicate significant differences between treatments (one-way ANOVA, *p* < 0.05). Scale bar = 1 cm.

**Figure 2 ijms-24-06380-f002:**
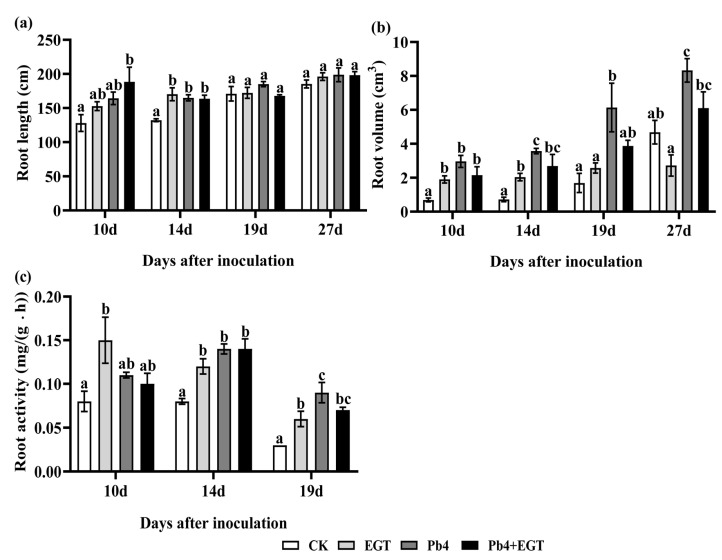
Effects of exogenous EGT on physiological indexes of Chinese cabbage root at different stages. (**a**) Root length (cm). (**b**) Root volume (cm^3^). (**c**) Root activity (cm^2^). Values are expressed in means ± SEM. Black letters indicate significant differences between treatments (one-way ANOVA, *p* < 0.05).

**Figure 3 ijms-24-06380-f003:**
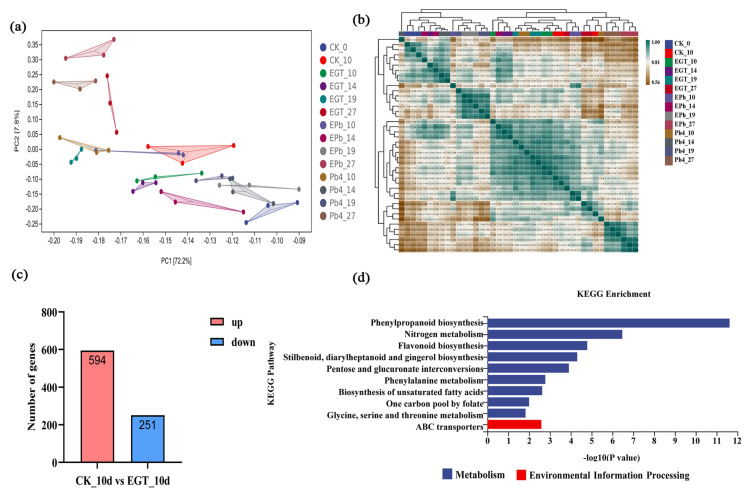
(**a**) PCA plots of the RNA-seq-derived transcriptomic responses. (**b**) The correlation coefficient analysis of transcriptome sample replicates. (**c**) Numbers of significantly differentially expressed genes between the control group and EGT group at 10 d. (**d**) KEGG enrichment analyses in comparison between the control group and the EGT group at 10 d. Numbers represent the significance of the number of genes significantly enriched in the corresponding pathway.

**Figure 4 ijms-24-06380-f004:**
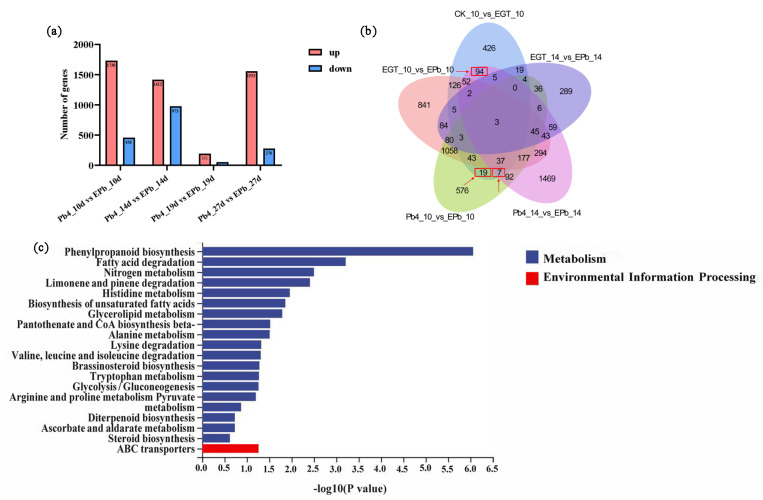
(**a**) Numbers of significantly differentially expressed genes between Pb4 group and EPb group at 10, 14, 19, and 27 dpi. (**b**) Venn diagram at ck-10d vs EGT-10d, EGT-10d vs EPb-10d, Pb4-10d vs EPb-10d, EGT-14d vs EPb-14d, and Pb4-14d vs EPb-14d. The 120 DEGs in the red box were further analyzed. (**c**) KEGG enrichment analysis of 120 DEGs.

**Figure 5 ijms-24-06380-f005:**
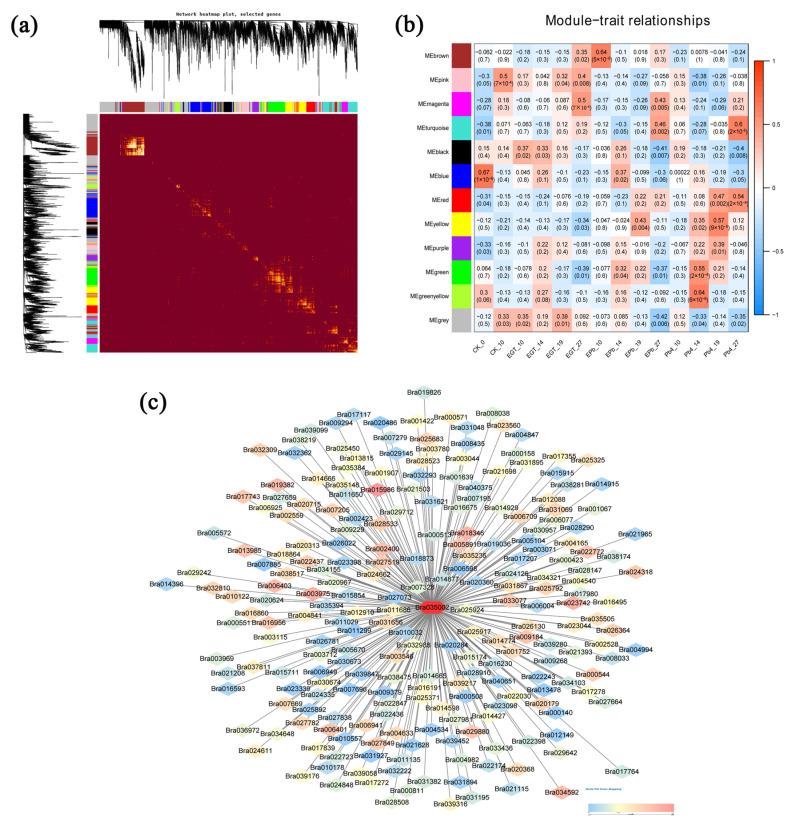
Weighted gene co-expression network analysis. (**a**) Heatmap plot of the topological overlap in the gene network. Each row and column correspond to a gene. A light color denotes low topological overlap. An increase in the topological overlap is reflected by an increase in the intensity of the red coloration. Dark squares along the diagonal correspond to modules. The gene dendrogram and module assignment are indicated along the left and top. (**b**) Module–traits relationships. The abscissa and the ordinate present the traits and the modules. The numbers represent the correlation between the modules and the traits. (+1 and −1, respectively, indicate strong positive and negative correlations between the modules and the traits). (**c**) The hub genes identified by the gene co-expression network of the “brown” module. The hub gene is a red circle in the center. 0 and 1 indicate strong correlation with the hub gene, with 0 in blue and 1 in red.

**Figure 6 ijms-24-06380-f006:**
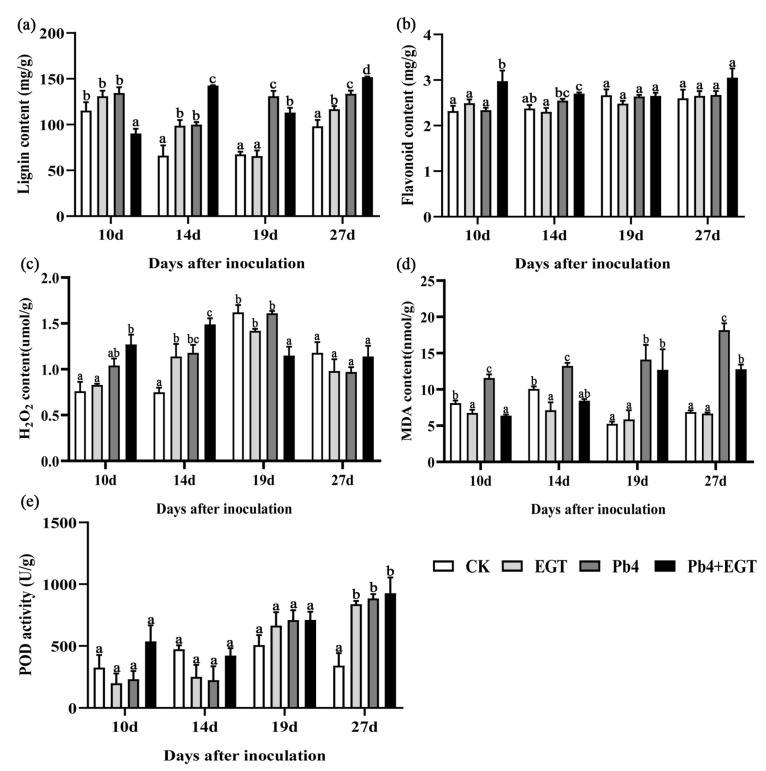
Detection of lignin content, total flavonoids, H_2_O_2_, MDA, and POD content. (**a**) lignin content, (**b**) total flavonoids content, (**c**) H_2_O_2_ content, (**d**) MDA content, (**e**) POD content. The contents of hydrogen peroxide (H_2_O_2_), malondialdehyde (MDA), and peroxidase (POD) in BJN3-2 were detected. Black letters indicate significant differences between treatments (one-way ANOVA, *p* < 0.05).

**Figure 7 ijms-24-06380-f007:**
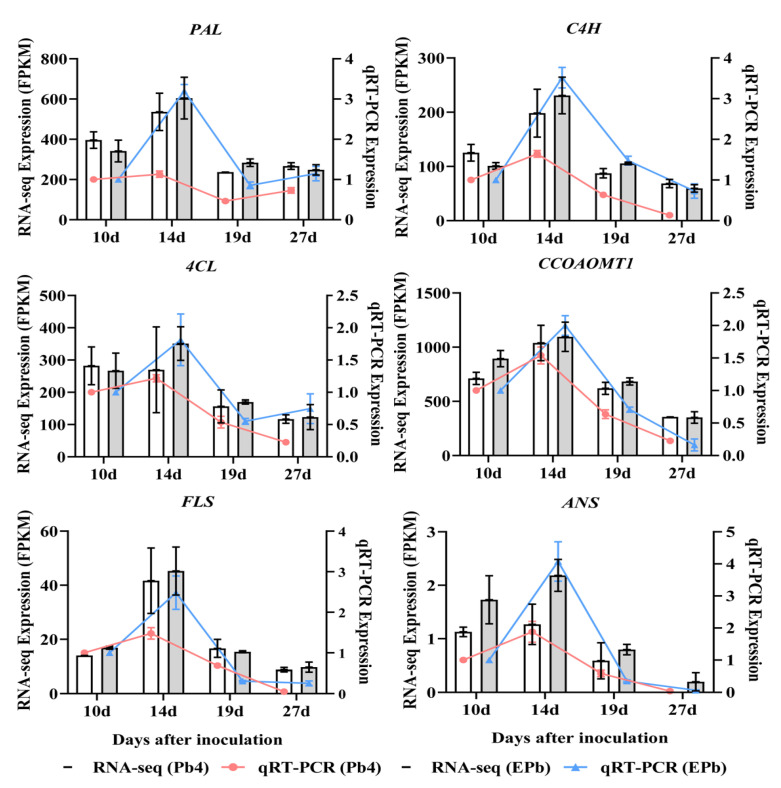
Verification of the phenylpropanoid pathway gene expression levels by qRT-PCR. (*PAL*, phenylalanine ammonia-lyase; *C4H*, transcinnamate 4-monooxygenase; *4CL*, 4-coumarate-CoA ligase; *CCOAOMT1*, caffeoyl-CoA O-methyltransferase; *FLS*, flavonol synthase; *ANS*, anthocyanidin synthase). The red solid line represents qRT-PCR for the Pb4 group; the blue solid line represents qRT-PCR for EPb group. The white bar chart represents RNA-seq expression for the Pb4 group; the grey bar chart represents RNA-seq expression for the EPb group (one-way ANOVA, *p* < 0.05).

## Data Availability

Not applicable.

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
