# Peer review of "Effects of Exogenous Ergothioneine on Brassica rapa Clubroot Development Revealed by Transcriptomic Analysis"

_ijms, 2023, doi:10.3390/ijms24076380_

Round 1

Reviewer 1 Report

This paper is a description of the mechanism by with ergothioneine (EGT) influences Clubroot disease condition in cruciferous plants. The paper is well written and provides novel findings on he role of EGT in plant disease suppression and plant growth. I propose that authors to consider the following:

1. In the conclusion, I would have loved the authors to provide a short opinion on how this can be applied in real time under field conditions as an alternative to the management of cruciferous clubroot diseases? How practical may it be for EGT to be applied in the soil e.g., close at the plant site or during planting? What about having lines with enhanced/optimized  EGT profiles. A note on the dimension of possible application of results, though not absolutely conclusive can significantly enrich the relevance of the study and point to an interesting direction going forward.

2. Figures and images need to be improved. It was a struggle for me to read text in nearly all the figures with tiny fonts except perhaps for figure 2, 6 and 7, although still some parts of a figure were poor but others better. Please improve the illustrations font sizes.

Author Response

Response to Reviewer 1 Comments

This paper is a description of the mechanism by with ergothioneine (EGT) influences Clubroot disease condition in cruciferous plants. The paper is well written and provides novel findings on the role of EGT in plant disease suppression and plant growth. I propose that authors to consider the following:

Point 1: In the conclusion, I would have loved the authors to provide a short opinion on how this can be applied in real time under field conditions as an alternative to the management of cruciferous clubroot diseases? How practical may it be for EGT to be applied in the soil e.g., close at the plant site or during planting? What about having lines with enhanced/optimized EGT profiles. A note on the dimension of possible application of results, though not absolutely conclusive can significantly enrich the relevance of the study and point to an interesting direction going forward.

Response: Sincerely thank you for your forward-looking comments. In this study, we applied exogenous EGT into the soil closed to the B. rapa plant roots cultured in the greenhouse. Our experiment with exogenous EGT was carried out in cell trays and has not yet been applied in the field yet. At present, we could only provide theoretical suggestions. In the future, we will continue to explore the application time and dosage of EGT in the field, and then provide practical suggestions for the prevention and control of cruciferous clubroot. As suggested by the reviewer, we revised the content in discussion section.

Point 2: Figures and images need to be improved. It was a struggle for me to read text in nearly all the figures with tiny fonts except perhaps for figure 2, 6 and 7, although still some parts of a figure were poor but others better. Please improve the illustrations font sizes.

Response: Thank you very much for the comments on figures and images. We have made a revision for all figures and figure legend.

Reviewer 2 Report

Dear Corresponding Author

I could not read your figures and charts. Therefore, you should improve the quality of the before of everything. After that, I can review your paper.

Regards

Author Response

Response to Reviewer 2 Comments

I could not read your figures and charts. Therefore, you should improve the quality of the before of everything. After that, I can review your paper.

Response: We are sorry for the trouble caused by the picture problem. We have improved the quality of all the pictures.

Reviewer 3 Report

LL259-267: Are there any studies on the action of EGT on spore germination of fungi or protists? If so, please cite them in the first paragraph of the DISCUSSION.

L335: EGT is the key compound in this paper; please list the manufacturer of EGT.

L143: Y-axis of Figure 2 (c) is wrong.

L206: “hug” may be “hub”.

Author Response

Response to Reviewer 3 Comments

Point 1: L259-267: Are there any studies on the action of EGT on spore germination of fungi or protists? If so, please cite them in the first paragraph of the DISCUSSION.

Response: Thanks for your suggestion. There are no studies on the action of EGT on spore germination of fungi or protists. However, some studies have shown that plant root exudates can promote the germination of resting spores of Plasmodiophora brassicae, and the root exudates of cruciferous host crops are more effective than those of non-host plants. We have revised and improved the manuscript in the corresponding place.

Point 2: L335: EGT is the key compound in this paper; please list the manufacturer of EGT.

Response: Thanks for your careful checks. We have listed the manufacturer of EGT in the manuscript line 390.

Point 3: L143: Y-axis of Figure 2 (c) is wrong.

Response: We have corrected it in the manuscript.

Point 4: L206: “hug” may be “hub”.

Response: We have corrected the “hug” into “hub”.

Reviewer 4 Report

Manuscript “Mechanism of the interaction between Plasmodiophora brassicae and Brassica rapa under exogenous ergothioneine treatment revealed by transcriptomic analysis” by Zhang et al. presents an interesting study about gene expression analysis of the resistance to a soil pathogen in Brassica rapa.

This manuscript presents a valuable study with production and breeding applications. However, this manuscript is very confused in some parts mainly in the description of the methodology and the experimental design. In addition, the manuscript presents important deficiencies mainly in the discussion of results. Phenotype results must be clarified. Validation of RNA-Seq through qPCR is not clear. For these reasons, this manuscript is ACCEPTABLE for publication in International Journal of Molecular Sciences after a major revision.

The major points for the REVISION of the manuscript are:

Around the whole manuscript, the complete name of genes should be in italics.

Objectives are very large. Authors must simplify the objectives not including any methodological reference.

Description of Results is very poor, mainly regarding phenotype analysis. The description of Brassica genotypes and their response is deficient. 

A new table with the response of the assayed genotypes must be included in the results section completing data from Figures 1 and 2. 

Quality of Figures 3 and 5 must be revised increasing font size. 

In Figure 7 authors should be incorporated the correlation coefficient between qPCR and RNA-Seq data in the assayed genes. 

Discussion section is very week. Authors must clarify the novelty of the obtained results in comparison with previous transcriptome data. It is necessary to transform RNA-Seq data in biological data, this is the nature of this manuscript. However, this biological data should be discussed in term of biological information adding some biological hypothesis clarifying the development of the peach fruit shape. The election of some genes for the monitoring of this process is also very important in the Discussion section. 

A new Conclusion section must be incorporated. Authors must indicate the main implications of these results from an agronomical and breeding point of view. 

Plant material must be completed describing main agronomic traits of the Brassica rapa genotypes assayed. 

Phenotyping must be clarified in the Methodology section. Evaluation of the Brassica rapa genotypes response toward Plasmodiophora brassicae must be clarified. At this moment information in sections 4.2, 4.3, 4.4 and 4.5 are not enough. 

From the methodology point of view, RNA-Seq and differential expression analysis are the key methodology used. RNA-Seq characteristics must be completed in section 4.6. In addition, biological and technical replicates must be described. Finally the origin of the tissue analysed in both assays (qPCR and RNA-Seq) must be clarified, flower, fruits, leaves, etc. 

qRT-PCR analysis should be also clarified indicating the nature of the assayed technical and biological samples. In my opinion RNA-Samples must come from a different assay, not exactly the same that the RNA-Seq assay. This question must be clarified. In addition, the selection of the candidate genes must be justified. If it is possible a higher number of genes to validate RNA-Seq data should increase the robustness of the experiment.

Author Response

Response to Reviewer 4 Comments

Point 1: Around the whole manuscript, the complete name of genes should be in italics.

Response: We have corrected the genes name in entire manuscript.

Point 2: Objectives are very large. Authors must simplify the objectives not including any methodological reference.

Response: Yes, we have not reached the research on the mechanism of exogenous ergothioneine to improve the resistance of Chinese cabbage to clubroot, so we have revised the title into “Effects of exogenous ergothioneine on Brassica rapa clubroot development revealed by transcriptomic analysis”. While, we didn’t include methodological reference when describing the objectives of the study.

Point 3: Description of Results is very poor, mainly regarding phenotype analysis. The description of Brassica genotypes and their response is deficient.

Response: Thanks for your comments but we are a little bit confused. This manuscript is about exogenous ergothioneine application to improve the resistance of Chinese cabbage to clubroot. The phenotype included disease rate, disease index and relative DNA quantity of P. brassicae which normally described for clubroot, were illustrated in the result part. The Brassica genotype is BJN3-2, a Chinese cabbage inbred line.

Point 4: A new table with the response of the assayed genotypes must be included in the results section completing data from Figures 1 and 2.

Response: Dear reviewer, we are a little bit confused about this question. Fig 1a is the observation of root phenotype of Chinese cabbage under different treatments. Fig 1b and 1c show the disease rate and disease index of clubroot of Chinese cabbage under different treatments. Fig 1d shows the quantitative detection of the content of P. brassicae in the roots of Chinese cabbage under different treatments. Fig 2 shows the measurement of root physiological indexes under different treatments, including root length, root volume and root activity.

Point 5: Quality of Figures 3 and 5 must be revised increasing font size.

Response: We have enlarged the font of all the figures in the manuscript.

Point 6: In Figure 7 authors should be incorporated the correlation coefficient between qPCR and RNA-Seq data in the assayed genes.

Response: The relationship between qPCR and RNA-seq data was mutually validated, and correlation coefficient was generally not used.

Point 7: Discussion section is very week. Authors must clarify the novelty of the obtained results in comparison with previous transcriptome data. It is necessary to transform RNA-Seq data in biological data, this is the nature of this manuscript. However, this biological data should be discussed in term of biological information adding some biological hypothesis clarifying the development of the peach fruit shape. The election of some genes for the monitoring of this process is also very important in the Discussion section.

Response: Thanks very much for your suggestion. However, we are really a little bit confused. The study is focus on Chinese cabbage clubroot, not development of the peach fruit shape. Anyhow, we revised the discussion part focusing one the effect of exogenous ergothioneine on Chinese cabbage clubroot.

Point 8: A new Conclusion section must be incorporated. Authors must indicate the main implications of these results from an agronomical and breeding point of view.

Response: In the future, we will continue to use ergothioneine to carry out field experiments and give suggestions on the prevention and control of clubroot. We have discussed in the discussion section.

Point 9: Plant material must be completed describing main agronomic traits of the Brassica rapa genotypes assayed. 

Response: We are a little confused about this question. The material is BJN3-2 which is a normal Chinese cabbage inbred line which has been described in Material and method section.

Point 10: Phenotyping must be clarified in the Methodology section. Evaluation of the Brassica rapa genotypes response toward P. brassicae must be clarified. At this moment information in sections 4.2, 4.3, 4.4 and 4.5 are not enough.

Response: We are really a little confused about this question.

Point 11: From the methodology point of view, RNA-Seq and differential expression analysis are the key methodology used. RNA-Seq characteristics must be completed in section 4.6. In addition, biological and technical replicates must be described. Finally the origin of the tissue analysed in both assays (qPCR and RNA-Seq) must be clarified, flower, fruits, leaves, etc.

Response: We have explained the relevant questions in 4.6 and 4.7. In this study, RNA-seq and qRT-PCR of roots which was generally for clubroot disease study were analyzed, not flower, fruits or leaves.

Point 12: qRT-PCR analysis should be also clarified indicating the nature of the assayed technical and biological samples. In my opinion RNA-Samples must come from a different assay, not exactly the same that the RNA-Seq assay. This question must be clarified. In addition, the selection of the candidate genes must be justified. If it is possible a higher number of genes to validate RNA-Seq data should increase the robustness of the experiment.

Response: RNA samples for qRT-PCR analysis was the same to the RNA-Seq assay.

We analyzed the differential genes and found that the differential genes were mainly enriched in the flavonoid pathway, so we selected the genes of the phenylpropanoid pathway for further quantitative verification.

Reviewer 5 Report

The authors present an interesting finding concerning the interaction between exogenous ergothioneine treatment and clubroot disease development. Through RNA-Seq analysis they link EGT to host secondary metabolism and the activation of the phenylpropanoid biosynthetic pathway. They go on to measure lignin and flavonoid levels, showing that the interaction between P. brassicae and EGT has the potential to increase levels of these metabolites. Furthermore, they show that malondialdehyde levels and lipid peroxidation may protected against by EGT. These findings will be of interest to clubroot researchers as well as those looking to test plant growth promoting chemicals. The manuscript is lacking several essential elements, there are no supplementary tables provided detailing the IDs of the differentially expressed genes or the various subsets and modules that have been highlighted. This means that despite taking a fairly detailed approach to mining their dataset the presentation of this data is entirely superficial and opaque.

The authors should upload their raw RNA-Seq data to SRA or some similar database in accordance with the policies of IJMS. 

It is interesting that exogenous EGT stimulated cabbage root growth, did this affect the growth of the plant above ground?

In Figure 1D, I think that a transformation of the data, such as log, may be useful. The significant differences at 10d are not viewable as presented. Furthermore, I wonder how the untransformed data can satisfy the requirements for normal distribution and equal variance that are assumed by ANOVA.

Figure 3d Are there numbers missing? The description says there are numbers indicating the number of transcripts significantly differentially expressed for each term but the only numbers are -log(10)p-values. Was a multiple testing correction used for the calculation of enriched GO terms?

I am a bit unclear about the selection of genes in Figure 4b, the intention seems to be to isolate the EGT response from the Pb response but there is no directionality in this analysis. Would a gene upregulated in response to EGT but downregulated in response to Pb not be of interest in this subsetting?

In the WGCNA analysis how was the "brown module" identified as a critical module for disease resistance? If the genes selected for the analysis in Fig 5 are are restricted to those with known functions can more informative labels be applied to the modules?

Figure 6 Are these genes supposed to be differentially expressed from the RNA-Seq data for the contrast shown EPB vs Pb? Because they don't look like they could be 2 fold differentially expressed. Why validate the expression of random pathway genes with qPCR? Are error bars missing from some of the qPCR data points?

285 "Therefore, we speculated that EGT might enhance the resistance of Chinese cabbage to clubroot by increasing POD activity" How do the authors square this claim with the lack of any significant difference in peroxidase activity between EPb and Pb?

I am interested in how the exogenous EGT was applied. In the methods it is described as being "injected". Was this injection to the plant or to the soil surrounding the roots. Was a needle used?

Regarding the RNA-seq data were paired ends or single reads captured? What read lengths were obtained? How many reads were obtained per sample. How many reads mapped to B. rapa genome? How many reads mapped to P. brassicae genes? Were Benjamini Hochberg adjusted p-values used in the identification of DEGs? No mention of this is made though it is the default option for DESEQ, if no MTC has been used then the number of DEGs must be revised to account for type i errors.

Minor points

1 I think that "Mechanism" is overstating the findings presented here. Certain processes are associated with the EGT P. brassicae interaction but no mechanism is tested

17 "phenylpropanoid" (phenylpropane is misused in several places)

19 change POD to peroxidase

32 "conditions, and its genetic diversity"

43 Ref 6 does not mention P. brassicae so this sentence should be rephrased to make it clear that changes in cell wall remodelling in general are being described

51 Ref 10 makes no mention of P. brassicae yet the reference is presented for a very specific statement about P. brassicae and cell wall lignification

76 "high speed railway hemoglobin"?

80 "ET" should be "EGT"

88 "phenylpropanoid"

109 "at 10"

Figure 1 and Figure 2: "means ± SD" but only upper error bar is visible

Figure 3b, the text on each correlation coordinate is very small, since it is colour coded would it look better without the text.

175 "phenylpropanoid"

194 "phenylpropanoid"

206 "hub"

209 "phenylpropanoid"

239 "phenylpropanoid"

240 "phenylpropanoid"

250 "phenylpropanoid"

247 What is the metric or basis for saying they exhibit "similar expression patterns"

290 "phenylpropanoid"

304 "phenylpropanoid"

313 "phenylpropanoid"

318 "phenylpropanoid"

383 DESEQ reference is missing

Author Response

Response to Reviewer 5 Comments

Point 1: The authors should upload their raw RNA-Seq data to SRA or some similar database in accordance with the policies of IJMS.

Response: Thanks for your suggestion. We have uploaded the raw RNA-Seq data into SRA repository, accession number is (PRJNA942197).

Point 2: It is interesting that exogenous EGT stimulated cabbage root growth, did this affect the growth of the plant above ground?

Response: We also observed the growth and development of the aboveground part of Chinese cabbage until the sampling day, but there was no significant difference.

Point 3: In Figure 1D, I think that a transformation of the data, such as log, may be useful. The significant differences at 10d are not viewable as presented. Furthermore, I wonder how the untransformed data can satisfy the requirements for normal distribution and equal variance that are assumed by ANOVA.

Response: Thanks to your suggestion, we have transformed data in the Figure 1d. The data in Figure 1d were analyzed by one-way ANOVA using SPSS, which we corrected in the manuscript. We were really sorry for our careless mistakes. Thank you for your reminder.

Point 4: Figure 3d Are there numbers missing? The description says there are numbers indicating the number of transcripts significantly differentially expressed for each term but the only numbers are -log(10)p-values. Was a multiple testing correction used for the calculation of enriched GO terms?

Response: There are no numbers are missing, we have misdescribed. In this paper, KEGG enrichment analysis is expressed by the significance of the number of differentially expressed genes. In the manuscript, we have corrected “Numbers represent the number of target transcripts significantly enriched in the corresponding pathway” to “Numbers represent the significance of the number of genes significantly enriched in the corresponding pathway”.

Point 5: I am a bit unclear about the selection of genes in Figure 4b, the intention seems to be to isolate the EGT response from the Pb response but there is no directionality in this analysis. Would a gene upregulated in response to EGT but downregulated in response to Pb not be of interest in this subsetting?

Response: Venn diagram can help us find the common and different points of the data set. We wanted to find out which metabolic pathways were enriched for the genes affected by ergothioneine at the early stage. We selected the common genes CK _ 10 vs EGT _ 10 and Pb4 _ 10 vs EPb _ 10, CK _ 10 vs EGT _ 10 and Pb4 _ 14 vs EPb _ 14, CK _ 10 vs EGT _ 10 and Pb4 _ 10 vs EPb _ 10 and Pb4 _ 14 vs EPb _ 14 for KEGG enrichment analysis.

Point 6: In the WGCNA analysis how was the "brown module" identified as a critical module for disease resistance? If the genes selected for the analysis in Fig 5 are restricted to those with known functions can more informative labels be applied to the modules?

Response: Each module has a module significance. The module significance represents the module itself to calculate the correlation with the trait (based on the sample). This correlation value is reflected in the heat map below. We selected the modules whose correlation was greater than 0.6 for analysis, and there were four modules (brown, turquoise, blue, and green yellow module). We want to find the module of the effect of ergothioneine on P. brassicae for further analysis. The analysis of the four selected modules showed that blue module was the control group at 0 d. The green yellow and turquoise modules were only inoculated with P. brassicae at 14 d and 27 d, respectively, and only P. brassicae was an influencing factor for these two modules. The brown module is an exogenous ergothioneine treatment followed by inoculation of P. brassicae, which can reflect the effect of ergothioneine, so we chose this module for further analysis. The genes of brown module were ranked according to the weight value, and Bra035002 was the highest. Annotation analysis of Bra035002 revealed that it encodes a CCR gene that plays a role in plant defense responses.

Point 7: Figure 6 Are these genes supposed to be differentially expressed from the RNA-Seq data for the contrast shown EPB vs Pb? Because they don't look like they could be 2 fold differentially expressed. Why validate the expression of random pathway genes with qPCR? Are error bars missing from some of the qPCR data points?

Response: In this part, we aim to verify the RNA sequencing results by qPCR analysis. Thus, we selected genes from phenylpropanoid pathway randomly, including both differentially expressed genes (ANS at 14 d) and non-significant differentially expressed ones. No error bars were missed, and there were several groups of genes expression with high concentration ratio.

Point 8: 285 "Therefore, we speculated that EGT might enhance the resistance of Chinese cabbage to clubroot by increasing POD activity" How do the authors square this claim with the lack of any significant difference in peroxidase activity between EPb and Pb?

Response: Thanks very much for your suggestion. It is true, though the POD activity in the EPb group was higher than that in the Pb4 group at the early stage of inoculation, but it was not significant. Therefore, we speculated that EGT might not be involved in enhancing the resistance of Chinese cabbage to clubroot by increasing POD activity. We have revised the in the discussion section.

Point 8: I am interested in how the exogenous EGT was applied. In the methods it is described as being "injected". Was this injection to the plant or to the soil surrounding the roots. Was a needle used?

Response: We used a medical syringe to inject EGT into the soil around the root of the Chinese cabbage. We have revised the corresponding content in material and method section.

Point 9: Regarding the RNA-seq data were paired ends or single reads captured? What read lengths were obtained? How many reads were obtained per sample. How many reads mapped to B. rapa genome? How many reads mapped to P. brassicae genes? Were Benjamini Hochberg adjusted p-values used in the identification of DEGs? No mention of this is made though it is the default option for DESEQ, if no MTC has been used then the number of DEGs must be revised to account for type i errors.

Response: RNA-seq data were paired ends captured. Sequencing read length is 150bp. The average raw read number was 46,270,556 for each sample. On average, 36,243,221 reads per sample were mapped to the B. rapa genome. We did not analysis and map the P. brassicae genes. The adjusted p-value was used for the identification of the DEG. We feel great thanks for your professional review work on our article.

Point 1: 1 I think that "Mechanism" is overstating the findings presented here. Certain processes are associated with the EGT P. brassicae interaction but no mechanism is tested.

Response: Thanks for your suggestion, we have changed the title of the manuscript to “Effects of exogenous ergothioneine on Brassica rapa clubroot development revealed by transcriptomic analysis”.

Point 2: 17 "phenylpropanoid" (phenylpropane is misused in several places)

Response: We were really sorry for our careless mistakes. Thank you for your reminding. We have corrected all "phenylpropane" to "phenylpropanoid"

Point 3:19 change POD to peroxidase

Response: We sincerely thank the reviewer for careful reading. We have corrected the “POD” into “peroxidase”.

Point 4: 32 "conditions, and its genetic diversity"

Response: We have changed as reviewer’s suggestion.

Point 5: 43 Ref 6 does not mention P. brassicae so this sentence should be rephrased to make it clear that changes in cell wall remodelling in general are being described.

Response: Thank you for your careful reading. There is a citation error here, which has been corrected in the manuscript.

Point 6: 51 Ref 10 makes no mention of P. brassicae yet the reference is presented for a very specific statement about P. brassicae and cell wall lignification

Response: We have corrected “High levels of cell wall lignification provide an effective defense for plants against P. brassicae” to “High levels of cell wall lignification provide an effective defense for plants against pathogens.”

Point 7: 76 "high speed railway hemoglobin"?

Response: We feel sorry for the mistakes. We have corrected "high speed railway hemoglobin" to “Methemoglobin”.

Point 8: 80 "ET" should be "EGT"

Response: We have corrected "ET" to "EGT".

Point 9: 88 "phenylpropanoid"

Response: We have corrected all "phenylpropane" to "phenylpropanoid".

Point 10: 109 "at 10"

Response: We have corrected the “at10” into “at 10”.

Point 11: Figure 1 and Figure 2: "means ± SD" but only upper error bar is visible

Response: Figure 1 and Figure 2 have been corrected in the manuscript.

Point 12: Figure 3b, the text on each correlation coordinate is very small, since it is colour coded would it look better without the text.

Response: As suggested by the reviewer, the text in Figure 3b has been removed.

Point 13-25:

175 "phenylpropanoid"

Response: We have corrected "phenylpropane" to "phenylpropanoid"

194 "phenylpropanoid"

Response: We have corrected "phenylpropane" to "phenylpropanoid"

206 "hub"

Response: We have corrected the “hug” into “hub”.

209 "phenylpropanoid"

Response: We have corrected "phenylpropane" to "phenylpropanoid"

239 "phenylpropanoid"

Response: We have corrected "phenylpropane" to "phenylpropanoid"

240 "phenylpropanoid"

Response: We have corrected "phenylpropane" to "phenylpropanoid"

250 "phenylpropanoid"

Response: We have corrected "phenylpropane" to "phenylpropanoid"

247 What is the metric or basis for saying they exhibit "similar expression patterns"

Response: It should be “similar expression trend”.

290 "phenylpropanoid"

Response: We have corrected "phenylpropane" to "phenylpropanoid".

304 "phenylpropanoid"

Response: We have corrected "phenylpropane" to "phenylpropanoid".

313 "phenylpropanoid"

Response: We have corrected "phenylpropane" to "phenylpropanoid".

318 "phenylpropanoid"

Response: We have corrected "phenylpropane" to "phenylpropanoid".

383 DESEQ reference is missing

Response: References have been added in the manuscript.

Round 2

Reviewer 2 Report

Dear Corresponding Author

For the moment I checked your paper and I think it is qualified enough to publication. Although, in some cases some English corrections are needed.

Regards

Author Response

We sincerely thank your valuable feedback, and we will try to improve the quality of the manuscript.

Reviewer 4 Report

Authors have revised correctly the manuscript 

Author Response

We feel great thanks for your professional review on our article.

Reviewer 5 Report

The authors have made some improvements to the presentation of the paper but essential elements are still missing. There is no supplemental table detailing which genes are differentially expressed. There should also be a supplemental table detailing the subset of DEGs selected in Figure 4b and a supplemental table detailing which genes are contained in each of the WGCNA modules in figure 5b.

In their response to my question about the biomass in the above-ground cabbage tissue the authors state that there was no difference with EGT treatment, does this lack of improvement in the yield of clubroot infected crops not undercut their proposition that EGT could be used to control clubroot?

In their response to my query about figure 6 the authors respond:

"Thus, we selected genes from phenylpropanoid pathway randomly, including both differentially expressed genes (ANS at 14 d) and non-significant differentially expressed ones."

However the methods still indicate that all the genes tested by qPCR were differentially expressed:

"To validate the RNA sequencing results, qRT-PCR was performed on nine selected DEGs in phenylpropanoid metabolic pathway"

Furthermore, they state that no error bars are missing from the qPCR data on figure 7 but there are several positions where no error bar can be distinguished. Perhaps adjusting the relative size of the error bars and the data point markers would help?

I observed that some basic information concerning the description of the RNA-Seq library preparation and sequencing was missing. They have responded but not changed the methods section or added the mapping statistics to the paper. The number of reads mapping will give readers and understanding of the quality and value of the experiment. They should detail how many reads were obtained in each run and how many could be mapped to the Brassica rapa genome as a supplemental table.

For the identification of DEGs the authors make it clear that the p-value threshold used was the BH adjusted p-value but this information has not been added to their description of the methods.

Minor points

125 “This is a figure” delete

In figure 1d, there is no P. brassicae DNA in the control CK plants does it make sense to include it as a column?

132 “Data was” OR “Values were”

Author Response

Response to Reviewer 5 Comments

Point 1: The authors have made some improvements to the presentation of the paper but essential elements are still missing. There is no supplemental table detailing which genes are differentially expressed. There should also be a supplemental table detailing the subset of DEGs selected in Figure 4b and a supplemental table detailing which genes are contained in each of the WGCNA modules in figure 5b.

Response: Thank you very much for your suggestions. The summary table of 120 DEGs has been added in the attachment Table S3. WGCNA module is Table S4 in the attachment.

Point 2: In their response to my question about the biomass in the above-ground cabbage tissue the authors state that there was no difference with EGT treatment, does this lack of improvement in the yield of clubroot infected crops not undercut their proposition that EGT could be used to control clubroot?

Response: Thanks very much for your comments. As shown in Fig 1, comparing with Pb4 group, DI and P. brassicae DNA quantity of plants treated with EGT (Pb4+EGT group) were significantly decreased at 14, 19 and 27d, though disease rate was 100% until 27d. There was no significant difference in biomass from 14d to 27d, only a slight yellowing phenomenon at the edge of the infected Chinese cabbage leaves in Pb4 group. P. brassicae infection limits the transport of nutrients and water in Chinese cabbage, and hinders its growth and development. When harvesting in the field, we can observe that the yield of Chinese cabbage infected by P. brassicae is significantly reduced. However, in this experiment, plants were grown in the plug trays for each treatment and cultured in the greenhouse. Thus, further field trials are needed to measuring the biomass during Chinese cabbage heading harvest time. We discussed about this point in discussion section.

Point 3: In their response to my query about figure 6 the authors respond: "Thus, we selected genes from phenylpropanoid pathway randomly, including both differentially expressed genes (ANS at 14 d) and non-significant differentially expressed ones." However the methods still indicate that all the genes tested by qPCR were differentially expressed: "To validate the RNA sequencing results, qRT-PCR was performed on nine selected DEGs in phenylpropanoid metabolic pathway"

Response: We are very sorry for our carelessness. We have corrected the “To validate the RNA sequencing results, qRT-PCR was performed on nine selected DEGs in phenylpropanoid metabolic pathway” into “To validate the RNA sequencing results, six genes were randomly selected from the phenylpropanoid metabolic pathway for qRT-PCR validation”.

Point 4: Furthermore, they state that no error bars are missing from the qPCR data on figure 7 but there are several positions where no error bar can be distinguished. Perhaps adjusting the relative size of the error bars and the data point markers would help?

Response: Thank you very much for the comments. The error bars are small because the errors among the three sets of data are small. We have tried as you suggested, and there is still no change. RNA-Seq and qRT-PCR data Table S4 has been added in the attachment.

Point 5: I observed that some basic information concerning the description of the RNA-Seq library preparation and sequencing was missing. They have responded but not changed the methods section or added the mapping statistics to the paper. The number of reads mapping will give readers and understanding of the quality and value of the experiment. They should detail how many reads were obtained in each run and how many could be mapped to the Brassica rapa genome as a supplemental table.

Response: Thank you for your suggestions. We have added relevant content to the manuscript in Results (line 157-161), Materials and Methods (line 420-433). Supplementary table1, 2 are attached.

Point 6: For the identification of DEGs the authors make it clear that the p-value threshold used was the BH adjusted p-value but this information has not been added to their description of the methods.

Response: Thanks for your careful checks. We have corrected in the manuscript line 454.

Minor points

Point 7:125 “This is a figure” delete

Response: Thanks for your careful checks. We have revised in the manuscript line 125.

Point 8:In figure 1d, there is no P. brassicae DNA in the control CK plants does it make sense to include it as a column?

Response: We think CK is useful as a blank control.

Point 9:132 “Data was” OR “Values were”

Response: Thanks for your suggestion. We have corrected the “Data was” into “Values were”.

Round 3

Reviewer 5 Report

I appreciate the authors efforts to respond to my comments on the previous version of the paper. I think that there is a communication problem though. In both of my previous reviews I have pointed out that there should be a supplemental table listing the DEGs that they have identified in their RNA-Seq experiments. Readers will want to know which genes are EGT responsive. In this current version supplemental tables are provided for the intersection of the DEGs and the WGCNA interactions which is great. However there is still no table listing the DEGs for each time-point and each contrast. One table with a tab for all the significant DEGs for, say, EGT 10d vs CK 10d and then another tab for the next contrast and so on. This is common feature of papers describing transcriptomic data, it would be really helpful for people interested to know whether their gene of interest responds to EGT and P. brassicae. Please consider adding this.

The table for the 120 intersecting genes is somewhat useful, I am curious how all the values are integers, are they read counts? FPKM values as they are labelled will not be whole integers. For a table of DEGs I think that the output of the DESeq analysis - the log2 ratio between conditions and the adjusted p-value for the contrast would be more immediately valuable information.

Line 605, the description of supplementary material should be updated

Author Response

Response to Reviewer 5 Comments

Point 1: I appreciate the authors efforts to respond to my comments on the previous version of the paper. I think that there is a communication problem though. In both of my previous reviews I have pointed out that there should be a supplemental table listing the DEGs that they have identified in their RNA-Seq experiments. Readers will want to know which genes are EGT responsive. In this current version supplemental tables are provided for the intersection of the DEGs and the WGCNA interactions which is great. However there is still no table listing the DEGs for each time-point and each contrast. One table with a tab for all the significant DEGs for, say, EGT 10d vs CK 10d and then another tab for the next contrast and so on. This is common feature of papers describing transcriptomic data, it would be really helpful for people interested to know whether their gene of interest responds to EGT and P. brassicae. Please consider adding this.

Response: We are very sorry for not understanding you well. Thank you for your patience. Your suggestion is very meaningful. We have additionally submitted a table containing all gene expression levels, a differentially expressed genes from different treatments, and CK _ 10 _ vs _ EGT _ 10, Pb4 _ 10 _ vs _ EPb _ 10, Pb4 _ 14 _ vs _ EPb _ 14, Pb4 _ 19 _ vs _ EPb _ 19 and Pb4 _ 27 _ vs _ EPb _ 27 differential genes. We can also see the changes of the different genes affected by EGT in other treatment groups.

Point 2: The table for the 120 intersecting genes is somewhat useful, I am curious how all the values are integers, are they read counts? FPKM values as they are labelled will not be whole integers. For a table of DEGs I think that the output of the DESeq analysis - the log2 ratio between conditions and the adjusted p-value for the contrast would be more immediately valuable information.

Response: Thanks for your comment. We have resubmitted Table S3 120 DEGs. DEG analysis was performed using DESeq to screen for DEGs with | log2foldchange | > 1 and false discovery rate (FDR) < 0.05

Point 3: Line 605, the description of supplementary material should be updated

Response: Thank you very much for your suggestion. We have updated the description of the supplementary material.
